# Analysis of the Pneumatic System Parameters of the Suction Cup Integrated with the Head for Harvesting Strawberry Fruit

**DOI:** 10.3390/s20164389

**Published:** 2020-08-06

**Authors:** Sławomir Kurpaska, Zygmunt Sobol, Norbert Pedryc, Tomasz Hebda, Piotr Nawara

**Affiliations:** Faculty of Production Engineering and Energetics, University of Agriculture in Krakow, ul. Balicka 116B, 30-149 Kraków, Poland; zygmunt.sobol@urk.edu.pl (Z.S.); norbert.pedryc@urk.edu.pl (N.P.); tomasz.hebda@urk.edu.pl (T.H.); piotr.nawara@urk.edu.pl (P.N.)

**Keywords:** pneumatic end effector, strawberries, damage

## Abstract

Fruit and vegetable harvest efficiency depends on the mechanization and automation of production. The available literature lacks the results of research on the applicability of pneumatic end effectors among grippers for the robotic harvesting of strawberries. To determine their practical applications, a series of tests was performed. They included the determination of the morphological indicators of the strawberry, fruit suction force, the real stress exerted by fruit suckers and the degree of fruit damage. The fruits’ morphological indicators included the relationships between the weight and geometrical dimensions of the tested fruit, the equivalent diameter, and the sphericity coefficient. The fruit suction force was determined on a stand equipped with a vacuum pump, and control and measurement instruments, as well as a MTS 2 testing machine. The necrosis caused by tissue damage to the fruits by suction cup adhesion was assessed by counting the necrosis surface areas using the LabView programme. The assessment of the necrosis was conducted immediately upon the test’s performance, after 24 and after 72h. The stress values were calculated by referring the values of the suction forces obtained to the surface of the suction cup face. The tests were carried out with three constructions of suction cups and three positions of suction cup faces on the fruits’ surface. The research shows that there is a possibility for using pneumatic suction cups in robotic picking heads. The experiments performed indicate that the types of suction cups constructions, and the zones and directions of the suction cups’ application to the fruit significantly affect the values of the suction forces and stresses affecting the fruit. The surface areas of the necrosis formed depend mainly on the time that elapses between the test and their assessment. The weight of strawberry fruit in the conducted experiment constituted from 13.6% to 23.1% of the average suction force.

## 1. Introduction

Due to the growing world population, the demand for fresh fruit and vegetables, and the limited amount of soil, higher agricultural productivity is increasingly required. The performance of agriculture is dependent on the mechanization and automation of production. According to recent data, in the US, strawberry production covers almost 20% of the world’s cultivated area [1]. The search for alternative cultivation methods in which the production process is fully automated is one of the directions of modern research. For example, Shamshiri [2] presented the results of using advanced technologies in a fully automated greenhouse (Green Factory). Agriculture provides a unique opportunity to develop robotic systems. The direction of research is the possibility of reducing production costs by automating its individual stages. Bechar and Vigneault [3] presented the results of research related to the use of robots in agriculture. The authors stated that the extensive use of the robotization of processes depends on the right sensors allowing the precise localization of the collected product, simple and reliable manipulators, and path planning algorithms.

An overview of modern sensor-based technologies, data analysis and robotics in field production (weed control, plant growth monitoring, robotic harvesting) was presented by Shamshiri [4]. The cost of labour related to crop production, harvesting and post-harvest activities account for the greater part of the total production costs, especially for special crops such as strawberries. The analysis related to strawberry harvest showed that, in China, the cost of strawberry harvest is over 25% of the total production cost [5], while in the USA in California, labour expenditure amounts to about 60% of the total operating costs [1].

Mu et al. [6] conducted tests on the collection of kiwi fruit, achieving over a 94% success rate, while the time of harvesting one fruit was 4 to 5 seconds. Birrell et al. [7] demonstrated the results of field tests of a constructed vision system that uses two integrated neural networks. The authors determined the success rate related to the location, meeting the quality standards of the collected products, and determined the necessary time for the complete harvest of lettuce. Shamshiri et al. [2], in their review work, made a synthetic overview of the current state and development prospects of robots in agriculture. Bloch et al. [8] reported the results of optimization of a robot dedicated to apple harvest, concluding that, in order to minimize the cost of harvesting, the components of the optimization procedure (the bush architecture, position of the robot arms and movement of the travelling platform) should be considered simultaneously. Pourdarbani et al. [9] presented the results of research related to the vision systems used in robots harvesting plums.

The first robots (Agribot) dedicated to fruit collection were presented by Ceres et al. [10], stating (based on the laboratory tests) in conclusion that the weakest point of its performance is the proper work of the gripper. Bachche et al. [11] presented an overview of robot design systems with vision systems and end effectors which have been applied.

## 2. Robotic Strawberry Harvest

For example, in the USA, nearly 70% of commercial companies in the strawberry marketing industry invest in robots for harvesting and packaging fresh strawberries [12]. However, efficient and reliable robotic strawberry picking turns out to be extremely difficult for several reasons. Among the most important factors can be distinguished the way of harvesting—the strawberries must be harvested in such a way as to eliminate or minimize bruising, the damage to the cover and the leakage of fruit juices—as well as the correct identification of the fruit. Ge et al. [13] presented the results of research on the use of the Deep Convolutional Neural Network to identify the degree of ripeness and shape deformation of harvested strawberries.

Automatic strawberry harvesting, also analyzed as part of agricultural robotics, generates additional specific problems [14]. The authors stated that strawberries, due to the texture of the surface of the fruit, have a lower light reflection coefficient, which makes their detection more difficult than apples, cucumbers, tomatoes or peppers. In addition, strawberries are more hidden inside the leaves and stems. Therefore, the location efficiency ratio of both the strawberry itself and its stalk is relatively low, e.g., the fruit location efficiency is less than 54% [15], and in the case of calyx it is successfully located only in about 60% of cases [16]. A good commercial practice is to protect harvested fruits from the sun and against warm wind and dust. It is required that, after harvesting, in accordance with the recommended practices, after a maximum of 2 hours, the fruits should be transferred to a cold store [17]. In addition to such features as appropriate size, colour, gloss, aroma and taste, strawberries are expected to have a desirable texture [17]. Chen and Opara [18] noted that the hardness of harvested fruits and vegetables influences not only their post-harvest treatment but also plays an important role in their transport. Aliasgarian et al. [19] analyzed damage to strawberries during their harvesting, post-harvest treatment and in the supply chain to points of sale, stating in conclusion that the largest share of damage occurs at the harvesting stage. As a result of the analysis, they also recommended the type of packaging and the arrangement of containers during transport. Akhijahcmi and Khodaei [20] showed in their research a strong reduction in resistance to damage formation of strawberry fruit along with an increase in their humidity. Korostynska et al. [21] presented the results of research on the use of a non-invasive electromagnetic method to assess the quality of strawberries, stating that it allows one to distinguish correctly the degree of ripeness of harvested fruits.

Kirk et al. [22] presented an analysis of the vision system used to identify the collected strawberries. Hayashi et al. [15] presented a commercial robot model for harvesting strawberries. The laboratory tests showed a success rate of almost 55%, and the picking time was approximately 9 seconds, which provided a movement speed of approximate 100 m/h. Qingchun [5] presented the results of their research on a prototype robot, in which the navigation of the robot’s position to the strawberry harvest was based on the measurement of Time Of Flight (TOF)—the so-called ultrasonic sonar. Hayashi [23] presented the results of an experimental robot consisting of a cylindrical manipulator, an end effector, a vision unit, a storage unit and a chassis. The robot was adapted to work at night. In another work of these authors, Hayashi et al. [24], it was found that the vision success rate is influenced by the declared degree of ripeness of the harvested fruit. The authors in the conducted tests determined the quantitative changes depending on the ripeness settings. Fountas et al. [25], in their review work, reviewed contemporary robots with a description of their equipment and a specification of their basic technical and operational parameters.

Thus, the further development of the cultivated area will depend on the development of robots which are adapted for fruit harvesting. Nawara et al. [26] presented the concept of strawberry harvester work in field crops. There are robot prototypes on the market, the most frequently cited of which are devices under the names Agrobot [27], CROO [28] and Rubion [29], and a robot prototype [30]. However, there is no detailed information on the effects of their use.

In addition to the vision system, the effects of a robotic harvesting system are also determined by the gripper which is used. 

## 3. Related Works Connected with the End-Effector

An overview of end effector constructions dedicated to fruit and vegetable harvesting was presented by Blanes et al. [31]. In addition to the construction used, the success of the gripper used in the robot is also determined by the physical properties of the products: the geometry, texture, size, friction coefficient, location of the center of gravity and susceptibility to mechanical damage, which significantly affect the manipulation of the robot arms.

Li et al. [32] presented the results of research on the finger gripper used for transplanting seedlings grown in peat substrate. Defterli et al. [33] reviewed the robots used in the cultivation and harvesting of strawberry fruit. Mohamed and Liu [34] analyzed the effects of using a gripper robot tip for the transplantation of seedlings, stating that the correct process is determined by soil moisture, which determines the value of the strength and efficiency of transplanted seedlings. Jiang et al. [35] developed a needle end effector (pneumatically driven) for the transplantation of plant seedlings, conducted tests of its effectiveness and determined the damage to the root ball. Davis et al. [36] presented the construction of a pneumatic end effector dedicated to handling delicate sliced fruits and vegetables. Zhang et al. [37] conducted operational tests in real conditions of a robot equipped with a finger gripper connected to the cutting device, intended for harvesting strawberries concluded that it is necessary to properly cultivate plants and select varieties with a regular shape of the fruit.

Chiu et al. [38] presented the results of a grip-type end-effector for greenhouse tomato harvesting. The gripper used consisted of four fingers controlled by electromagnets, the inner part of which was covered with a sponge. Wang et al. [39] presented the results of research on a tomato harvesting gripper, which uses a mechanism of sliding jaws covered with gas cushions. Pettersson et al. [40] described an innovative end effector, the principle of operation of which is to use the magnetic properties of the fluid placed in the gripper’s fingers. Dimeas et al. [41] presented the results of tests on a three-finger form gripper cooperating with the robot arm. Chen et al. [42] analyzed the selected parameters of fruit contact with the fingers of human hands when sorting fruit. Mu et al. [6] presented the effects of a two-finger gripper for harvesting kiwi, which uses pressure sensors to minimize damage. De Preter et al. [43] presented the effects of a strawberry harvesting robot in which a finger gripper was installed. 

As can be seen from the cited review, the issue of robotic fruit or vegetable crop harvesting is a modern research trend implemented in various scientific centers. These works concern various agricultural products, including strawberry fruit. An end effector is an extremely important element of the robot, and the one with a vacuum gripper can be distinguished among the types used.

Based on the available research results presented in scientific papers, there is a lack of information related to the construction of the suction cup and the required negative pressure, and analysis of the extent to which the applied negative pressure is destructive to strawberry fruit. Hence, the main purpose of this work is to determine the operating parameters of the vacuum end effector in terms of the efficiency of the suction of strawberry fruit, taking into account the resulting mechanical deformations.

## 4. Research Methodology

### 4.1. Determination of the Suction Power of Strawberry Fruit

The suction force of the fruit by the tested suction cups was determined on the stand shown in Figure 1. The stand was equipped with vacuum pump 8, driven by electric motor 9. By means of valve 6, the flow and negative pressure prevailing in the suction cup work system were regulated. Vacuum gauge 7 was used to monitor the vacuum parameters. The maximum operating vacuum was 5 kPa, and the air flow velocity, depending on the degree of sealing between the surface of the suction cup face and the fruit was 2–10 m/s. The sucked fruit was connected to measuring head 3 of the MTS 2 (Material Testing Systems) testing machine by means of a ‘weightless’ flexible, non-flexible connector. The course of changes in the force value read through strain gauge 3 as a function of the displacement of the head was recorded in computer programme 1. The maximum force measured by the head was used to analyze the maximum fruit suction force by the suction cup.

At the measuring stand, the sensors used were characterized by the following measuring accuracy: (a) vacuum gauge: range -1 to 5 bar, reading accuracy 0.1 bar; (b) speed measurement: a wind probe with a range of 0 to 40 m/s and an accuracy of 0.2 m/s; the measurement was carried out using a head compatible with MTS 2, with a measuring range of 0–25N and a reading accuracy of 0.001N.

The tests were carried out on three types of suction cups (Table 1). The suction cups were made of silicone. The tested suction cups had a round face, while the shape of the suction cup chamber was conical, in the form of an inverted truncated cone. The feed hole’s diameter was 10 mm. Two medium deep (15 mm) and one shallow (10 mm) suction cups were used. Among the deep suction cups, one larger example, with an outer surface diameter of 43 mm, and another smaller example, with a diameter of 25 mm, were tested. The number and shape of the suction cups were selected on the basis of the previous analysis of the morphological characteristics of the strawberry fruit. As a consequence, two ventricular shapes were adopted for the research—medium-deep and shallow (types 1 and 2). As a result, the influence of the inner walls of the suction cup on the ability to adhere to fruit was checked (suction cup sealing to the surface of the fruit). Within the medium-deep suction cups, two dimensions of the seat surface (two diameters) were adopted for the tests—types 1 and 3. This choice made it possible to analyze the value of the suction forces (at constant negative pressure) and the stress values at the interface between the fruits’ surface and the surface of the suction cup. The assessment of these parameters illustrates the effectiveness of fruit suction (effectiveness of the suction cups) and the possible formation of mechanical damage to the tissue on their surfaces.

The tests were carried out within the following zones and directions of suction cup placement (Figure 2):The top zone of the fruit, 45° to the axis (contained between the apical part and the part of the fruit attachment);The central zone of the fruit, 90° to the axis (contained between the apical part and the part of the fruit attachment);The pedicel zone of the fruit, 135° to the axis (contained between the apical part and the part of the fruit attachment).

These features were selected for the analysis because of their decisive importance in adjusting the technical parameters of the suction cups, i.e., the shape, dimensions and operating parameters of the suction cups (negative pressure values). By describing the shape, through the mutual relations of their basic dimensions, fruits can be classified into the groups ‘shortened’ or ‘elongated’, and from this, one can infer the size and shape of the individual fruit zones, i.e., middle, apical and pedunculated. Furthermore, from the level of correlation between the weight and the individual basic dimensions, it can be inferred that the tissue mass is distributed within the fruit, and this may suggest some recommendations regarding the shape of the cup, its size, and the place of its application to the surface of the fruit (fruit zone).

### 4.2. Morphological Indicators of Strawberry Fruit

In order to thoroughly analyze the process of the fruit suction by the suction cup, the most important morphological features of the examined fruit and the relations between them were determined. For the studied fruit, the morphological indices of strawberry fruit were calculated [19]:
the equivalent diameter and sphericity coefficient (D_d_):(1)$$D_{d}={\left(d^{2}\cdot c\right)}^{0.33}$$the coefficient of sphericity (Φ):(2)$$\Phi =\frac{{\left(c\cdot d^{2}\right)}^{0.33}}{c}$$

The strawberry diameter was calculated as the arithmetic mean of the dimensions a and b (Figure 2), where a is the smaller dimension from the cross section and b is the larger dimension:(3)$$d=\frac{a+b}{2}$$
where c is the dimension between the apex (the top zone of the fruit) and the fruit attachment (mm).

The geometrical dimensions were measured using an electronic caliper with a reading accuracy of 0.01mm, while the strawberry fruit’s weight was measured using laboratory scales with a range of 0–300g and a measurement accuracy of 0.0001g. 

The correlation between the basic fruit dimensions (a, b, c) and the equivalent diameter was examined (D_d_), as well as the coefficient of sphericity (Φ) and the weight of the fruit.

### 4.3. Examination of the Degree of Damage to the Strawberry Fruit (Surface Necrosis)

The degree of the damage to the fruit as a result of the stress resulting from the impact of the suction cups was examined by analyzing the necrosis formed after tissue damage. The tests were carried out by measuring the surface area of the tissue discoloration (the necrosis formed) in the places where the impact of the sucker on the fruit tissue occurred. The surface area of necrosis was counted using the LabView programme. The tests were carried out in three terms: 1) directly after the suction cup has been applied to the tissue, 2) 24 hours after the suction test, and 3) 72 hours after the test. Photos of the strawberries created in the LabView program without exposure and after the action of the suction cups were used for the necrosis area determination. The necrosis area was defined as the difference of pixel colors between the strawberry without the action of the suction cup and the color of the pixels after the action of the suction cup for the area treated with the suction cup. Subsequently, the necrosis area was calculated from the pixels per unit area [mm^2^].

### 4.4. The Study of the Actual Stress Exerted on the Strawberry Fruit

In order to determine the stress S affecting the fruit surface it is important to provide the measurements of the suction cup surface A, which acts on the fruit with the suction force F_ch_. Calculations were made by dividing the value of the suction force F_ch_ by contact of the suction cup surface with the fruit surface (A). The simplicity of the calculations were made due to the fact that the actual contact surface of the suction cup face with the fruit surface area cannot be easily determined. It was assumed that the entire surface of the suction cup face actively participates in the impact on the fruit:(4)$$S=\frac{F_{ch}}{A}$$
where F_ch_ in the suction force of the fruit to the suction cup N, and A is the surface area of the suction cup face, m^2^. 

The obtained test results were analysed using the STATISTICA 13.3 programme at the assumed significance level of α = 0.05. The normality of distribution (Shapiro–Wilk test) and homogeneity of variance in the samples (Levenea test) were tested. An analysis of variance with Duncan’s test was applied. A regression analysis was conducted, searching for substantively justified correlations between the tested quantities.

## 5. Test Results

### 5.1. Morphological Characteristics of the Examined Fruits

Strawberries of the San Andreas variety were used for all performance tests. The analysis was conducted for 75 pieces of strawberry fruit. The average relative error in the measurement of morphological parameters ranged from 0.02 to 0.03%, while the measurement of the mass was 0.0003%. The maximum absolute error of the calculation of the equivalent diameter and the sphericity coefficient (calculated using the logarithmic method, assuming additivity partial errors) was 0.0002 mm (for diameter) and 0.001 (for sphericity coefficient).

In order to characterize the selected quantities and morphological structure of fruits in a comprehensive way, a weight regression analysis (0.21–0.42 N) on the basic fruit dimensions was performed. The impact of the basic dimensions—a, b, c—of fruit on their weight, turned out were statistically significant in relation to all dimensions. The highest degree of the fruit weight was positively correlated with dimension a (33.9–63.3 mm), for which r^2^ = 0.85 (Figure 3). To a lesser extent, the fruit weight was positively correlated with dimension b (31.2–52.7 mm), for which r^2^ = 0.38 (Figure 4), and dimension c (29.8–42.7), for which r^2^ = 0.27 (Figure 5).

Equivalent diameters and sphericity coefficients were calculated for the examined fruits. Equivalent diameter values D_d_, for the tested fruits, ranged from 32.2 mm to 47.4 mm, and the sphericity coefficients Ø ranged from 0.936 to 1.269. The regression analysis shows that the weight of the fruit depended, in a statistically significant manner, on the equivalent diameter (Figure 6) and the sphericity coefficient (Figure 7). The fruit weight was positively correlated with the equivalent diameter D_d_, for which the linear regression coefficient was r^2^ = 0.77 and coefficient of sphericity Ø was r^2^ = 0.27. 

### 5.2. Suction Force of Strawberry Fruits

The analysis of variance in the double classification demonstrated that the value of the suction force of strawberry fruit was statistically significantly affected by the types of suction cups, and the application zones and directions of the suction cups on the fruit. The interaction between the adopted factors proved to be statistically significant (Table 2).

The system of homogeneous groups (determined using Duncan’s test) shows that the highest values of strawberry fruit suction forces (Table 3, Figure 8) were obtained with the use of suction cups 1 and 3, 2.38 and 2.26 N, respectively, belonging to one homogeneous group. A separate group of homogeneous values of the suction force of the fruit were obtained for suction cup 2, for which the average value of the suction force was 1.86 N. The obtained test results show that the highest values of the suction force were obtained using suction cups in which the depth (the distance between the surface of the face and the bottom of the suction cup chamber) was 15 mm, regardless of different diameters of the face (Table 1 and Table 3).

Based on the obtained results, it can be concluded that the suction force of the fruit depends on the distance between the surface of the face and the bottom of the suction cup chamber (dimension h_s_, Table 1). The greater depth of the suction cup chamber allows for better adaptation of the suction cup face to the fruit surface, better sealing of the suction cup against the fruit, and thus obtaining higher suction forces. When the suction cup chamber is flattened, the fruit may come into contact with the inner wall of the suction cup chamber instead of the face, which is probably the consequence of a smaller degree of sealing of the suction cup with the fruit, and lower suction values. 

The use of different zones and directions of suction cup application resulted in a statistically significant differentiation of the average value of the suction force (Table 4, Figure 9). The average values of the forces obtained in relation to the zones and directions of application of the suction cups were divided into three homogeneous groups. The smallest average value of the suction force, 1.60 N, was obtained in zone 1 for vertical position 45° in relation to the line direction between the apical part and the trailer part of the fruit (group 1, Table 4). The highest average value of 2.72 N suction force was obtained in the 2^nd^ zone, and the application direction was to the middle zone of the fruit, 90° in relation to the axis (contained between the apical part and the part of the fruit attachment) (group 3, Table 4). In zone and direction 3, i.e., the pedicel fruit zone, 135^0^ relative to the axis (contained between the apical part and the part of the fruit attachment), the average value of the suction force was 2.16 N (group 2, Table 4).

The diversity of the suction force value depending on the zones and directions of suction cup application results mainly from the morphological structure of the fruit. Obtaining the smallest suction force at the top of the fruit is associated with the occurrence of the small values of the radius of curvature of the fruit (decreasing along the course of the measurement site towards the apical part of the fruit), occurring transversely in relation to the axis contained between the apical part and the part of the fruit attachment. A similar occurrence of small values of the fruit shape radius can be indicated at its base, with their direction being reversed in relation to the apical part. The result of the occurrence of a small fruit surface is a worse fitting of the suction cup face surfaces, and their worse sealing with the fruit, which causes us to obtain lower values of suction forces. 

In the middle part of the fruit, the curvature of the fruit surface is described by radiuses of high values, occurring in both directions in relation to the axis of symmetry. In this zone, a suction cup applied perpendicularly to the axis, connecting the top part of the fruit with the attachment part, was probably best sealed, and the greatest suction force was obtained with this combination. 

Considering the simultaneous interaction among the types of suction cups adopted for testing, as well as the zones and directions of the suction cup application, it should be stated that the most beneficial solutions in which the highest suction efficiency was obtained were achieved in combinations of 1 and 3 types of suction cups, zone 2 and perpendicular direction of application. For obtained parameters the average value the suction force was 3.03 N (group 4). The least favorable solution was obtained mainly in zone 1, and the direction of application of the suction cups regardless of the type of suction cups (group 1). For this homogeneous group, the average value of the suction force was from 1.56 to 1.87 N (Table 5, Figure 10).

The suction force values obtained were compared with the weight of the tested strawberry fruit. The average weight of the tested strawberry fruit was 0.37 N, and accounted for 13.6% of the value of the suction force (2.72 N) in the second zone and the application direction to the middle zone of the fruit, 90° in relation to the axis (contained between the apical part and the part of the fruit attachment), up to 23.1% values of the suction force (1.60 N) in zone 1 and the direction of application to the peak fruit zone, 45° in relation to the axis (contained between the apical part and the part of the fruit attachment) (Figure 11).

### 5.3. Stresses Occurring at the Contact Point of the Fruit with the Suction Cup

Analysis of variance in the double classification showed that the value of stress occurring at the point of contact of the fruit with the suction cup was statistically significantly, and was affected by the type of the suction cups, and the zones and directions of application of the suction cups to the fruit. The interaction between the adopted factors turned out to be statistically significant (Table 6).

The highest stress value affecting fruits occurred when suction cup 3 was used (Figure 12, Table 7). This suction cup has a conical chamber shape, in the form of an inverted truncated cone with an outer diameter of Ø = 25 mm and an internal diameter of Ø = 23 mm. Face surface A is 75.4 mm^2^ (Table 1). It differed substantially in the values of the face diameters from the other two suction cups used in the experiment (1 and 2), for which the surface areas of the face were: suction cup 1: A = 257.5 mm^2^; suction cup 2: 251.2 mm^2^. These significant differences in the stress values generated by suction cup 3 (0.0300 MPa) and the other suction cups (suction cup 1: 0.0092 MPa; 2: 0.0074 MPa) result from the relationship between the entrustments of these suction cups’ faces, which is about 1: 3.4 (Figure 12, Table 7). 

The statistically significant difference in stress that occurs as a result of the use of suction cups 1 and 2 (Figure 12, Table 6 and Table 7) is caused, in turn, by a significant difference in the value of the suction forces, because the surface areas of the face of suction cups 1 and 2 are comparable with each other. However, the average value of the suction force for suction cup 1 was 2.38 N, and for suction cup 2 was 1.86 N, and these values were statistically differentiated.

Stress values in the zones and directions of interaction of the suction cups on the fruit tissue were statistically significantly differentiated (Table 6 and Table 8, Figure 13). The lowest stress value was recorded in zone 1 and the direction of application of the suction cups 0.0112 MPa; the higher value was in 3 (0.0152 MPa), and the highest in the second zone (0.0201 MPa) (Table 8, Figure 13). These changes were closely related to the changes in the suction forces of the fruit by the suction cups occurring in individual impact zones. The course of changes in the values of the forces and stresses in relation to the zones and directions of the application of the suction cups was the same.

The obtained results of the stress tests affecting the fruit tissue resulting from the action of suction cup faces in the fruit suction (gripping) tests were compared with the results of the fruit penetration tests of 10 dessert varieties and two processing varieties (47). The loading element was a cylindrical shaft with a diameter of 6 mm. In the mentioned studies, the penetration tests were carried out on a machine comparable to the MTS 2 testing machine INSTRON 4301. The authors obtained the values of the destructive stress in the range of 0.0654–0.1011 MPa. In the analyzed results (Table 9), affects the stress values of the fruit tissue due to the impact of suction cup surfaces. The stress value was ranged from 0.0062 MPa for the type 2 of suction cups and zone 1 to 0.0402 MPa for the type 3 of suction cups and zone 2 (Table 9, Figure 14). The stress values of the suction cups accounted for 10–40% of the stress values of the penetration test. Nevertheless, the tissue tests performed after the fruit suction tests demonstrated that, as a result of this operation (with given assumed technical and operational parameters), tissue damage occurs. Necrosis is formed on the surface of the damaged tissues.

### 5.4. Necrosis Formed after Tissue Damage Due to the Suction Cups’ Impact

The analysis of variance in the double classification showed that the necrosis test date had a statistically significant effect on the value of the necrosis area. On the other hand, the type of suction cups used did not statistically affect the size tested (Figure 15). Additionally, the interaction between the adopted factors turned out to be insignificant (Table 10).

The average value of the surface of the necrosis determined directly after loading the fruit (term 1) (after being sucked by the suction cup) was 26.40 mm^2^; after 24 h (term 2) it was 28.43 mm^2^, and after 72 h (term 3) it was 31.79 mm^2^ (Figure 16, Table 11). This course of changes in the growth of the necrosis surface with the passage of time, from the moment of tissue damage, is logical, and results from the gradual disclosure of the colour of oxidized damaged cells. The Duncan test (Table 11) demonstrated that the adopted deadlines for assessing the surface area of the damaged fruit tissues statistically significantly affects the size of the necrosis field formed, dividing the obtained values into three separate homogeneous groups (Table 11).

The average values of necrosis areas within the type of suction cups and the necrosis testing term, as well as the visualizations of selected cases, are presented in Table 12. Analyzing in detail the values of the surface areas of the examined necrosis, it can be stated that they are in the range from 25.84–27.92 mm^2^ (homogeneous group 2 and first test term for all types of suction cups) to 30.96—32.76 mm^2^ (homogeneous group 3, which covers the third test date and all types of suction cups) (Figure 17, Table 13). 

## 6. Discussion

The tests performed clearly show that it is possible to use pneumatic suction cups in the picking heads. However, there is a need to choose their construction correctly. The tests that were made on standard silicone suction cups demonstrated that the smallest increase in the damaged surface was observed for the type 1 suction cup, in which, after 72 hours, the necrosis area increased by about 18.2%. In turn, for suction cup type 3, this increase was over 23%. The observed increasing in the necrosis surface were conducted with the storage of the harvested fruit, in which only the humidity and air temperature were maintained at the recommended level. According to the available research, the storage stability can be increased by treating the fruit with pulsed light [44] and an increased concentration of CO_2_ in the storage chamber [45]. However, it should be taken into account that there is diversity among the mechanical properties between strawberry varieties [46]. A separate issue directly related to the possibility of using pneumatic suction cups in end effectors is the control of the movable end effector’s robotic arm. As research has shown, the greatest suction force was obtained when the suction cup was directed to the central part of the fruit—perpendicular to its axis of symmetry. An important issue in use of pneumatic suction cups is to answer the question: Is the generated stress (coming from the vacuum pump) is not so destructive to the fruit tissue. To prevent damages for the fruit texture, the value of the suction force was compared with the weight of the fruit. The conducted research has shown that generated stresses were twice times lower than the destructive stresses. The research conducted showed clearly that these stresses are on average two times smaller than destructive stresses. This conclusion was formulated on the basis of the comparison of the obtained results with research results [46,47]. It was also found that the suction force is from four to nearly eight times greater than the weight of the fruit, which guarantees the stability of fruit adhesion on the gripper’s movement track (from the plant to the collecting container). Summarizing the obtained results, the possibilities of using pneumatic suction cups were confirmed and could be use in gripper tips. 

## 7. Conclusions

The types of suction cups used have a statistically significant effect on the value of the suction force. The highest value of the fruit suction force was obtained for the suction cups designated 1 (2.38 N) and 3 (2.26 N).The zones and directions of the suction cup application have a statistically significant impact on the value of the suction force. The highest value of the fruit suction force (2.72 N) was obtained for zone and direction 2 (the middle zone of the fruit, 90° in relation to the axis (contained between the apical part and the part of the fruit attachment)).The most advantageous solution, in the context of fruit suction efficiency, obtaining the highest suction force (3.03 N) among the tested solutions, is the use of type 1 or 3 suction cups in the second zone and direction of suction cup application.The average weight of strawberry fruit in the experiment was from 13.6 to 23.1% of the average suction force. This suggests that the vacuum system under study can perform operations to extract the fruit from the bush, transport them after cutting the stem, or even detach the fruit from the stem in the case of fruit harvesting for processing.The value of the stress acting on the fruit coming from the suction cup pressure depends on the type of suction cups and the zones and directions of the application of the suction cups to the fruit.The highest generated stress value (0.0300 MPa) were occurred for suction cup 3 and were three times smaller than other suction cups used in the experiment, i.e. cup 1 (0.0092 MPa) and cup 2 (0.0074 MPa).

## 8. Patents

The concept of the working section of the strawberry harvester has been reported as a utility model.

## Figures and Tables

**Figure 1 sensors-20-04389-f001:**
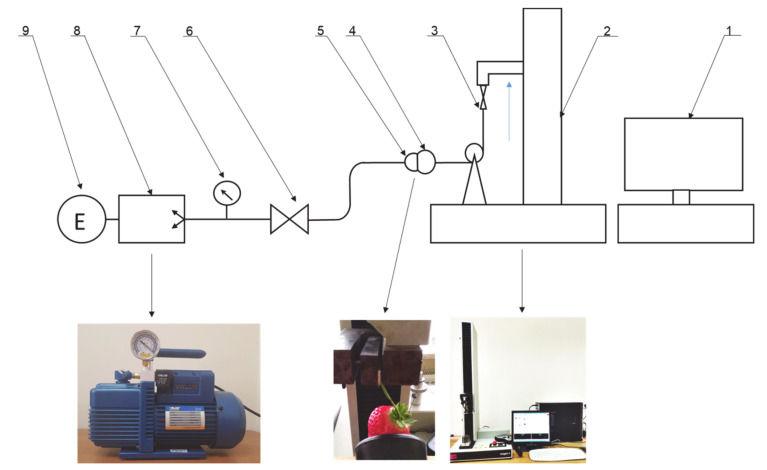
Diagram of the measuring stand for determining the maximum suction force of strawberry fruit. 1: computer, 2: MTS 2 testing machine, 3: load cell strain gauge, 4: strawberry fruit, 5: suction cup, 6: valve, 7: vacuum gauge, 8: vacuum pump, 9: motor.

**Figure 2 sensors-20-04389-f002:**
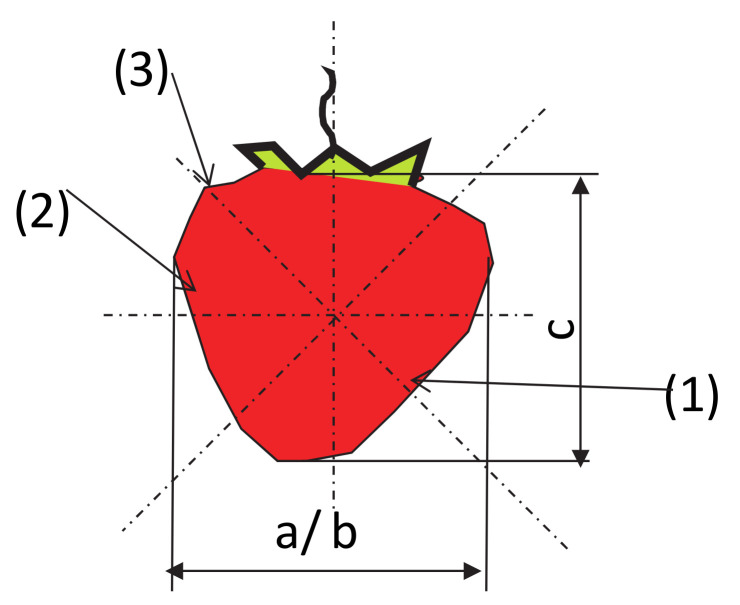
Zones and directions for applying the suction cups to the fruit during testing—1: the top zone, 2: the central zone, 3: the pedicel zone.

**Figure 3 sensors-20-04389-f003:**
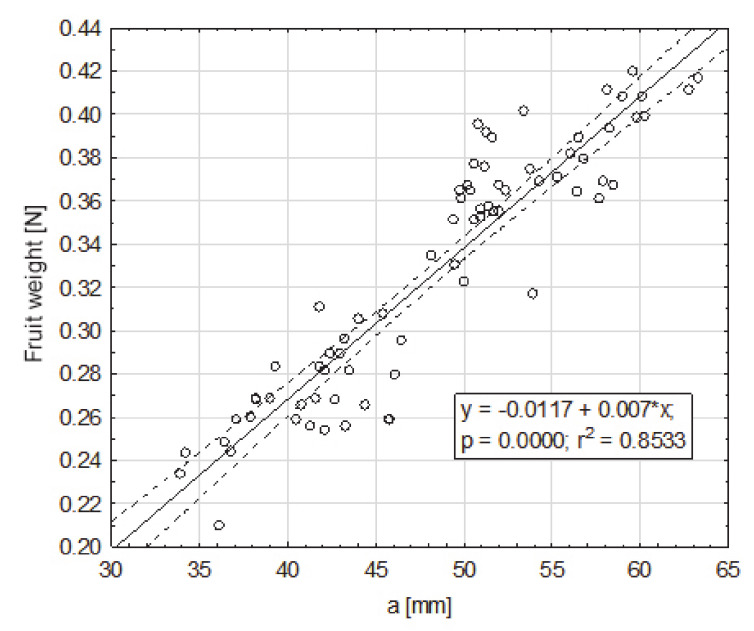
Dependence of the weight of the fruit on dimension a (mm), smaller, from the cross-section to the axis extending between the apical part and the part of the fruit attachment, in the maximum place.

**Figure 4 sensors-20-04389-f004:**
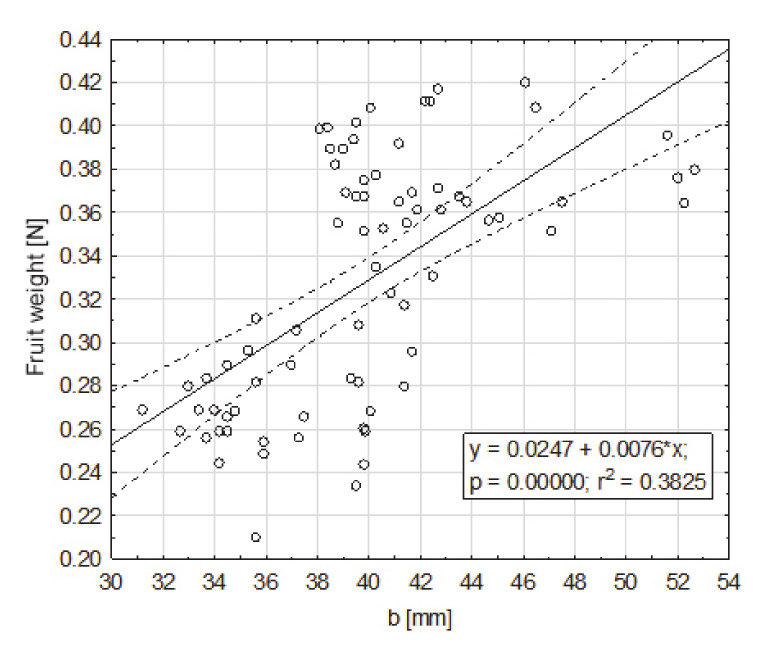
Dependence of the fruit weight on dimension b (mm), greater, from the cross section to the axis contained between the apical part and the part of the fruit attachment, at the maximum place.

**Figure 5 sensors-20-04389-f005:**
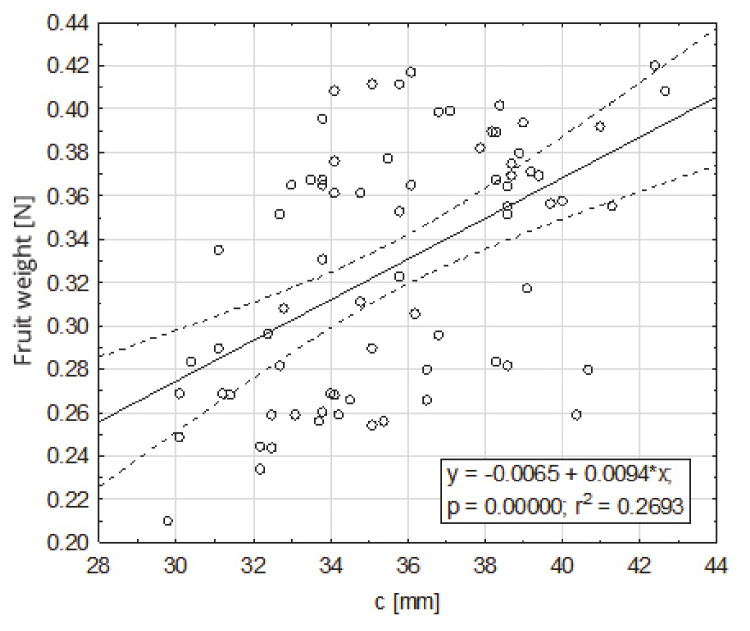
Dependence of the fruit weight on dimension c (mm), contained between the top of the fruit and the fruit attachment.

**Figure 6 sensors-20-04389-f006:**
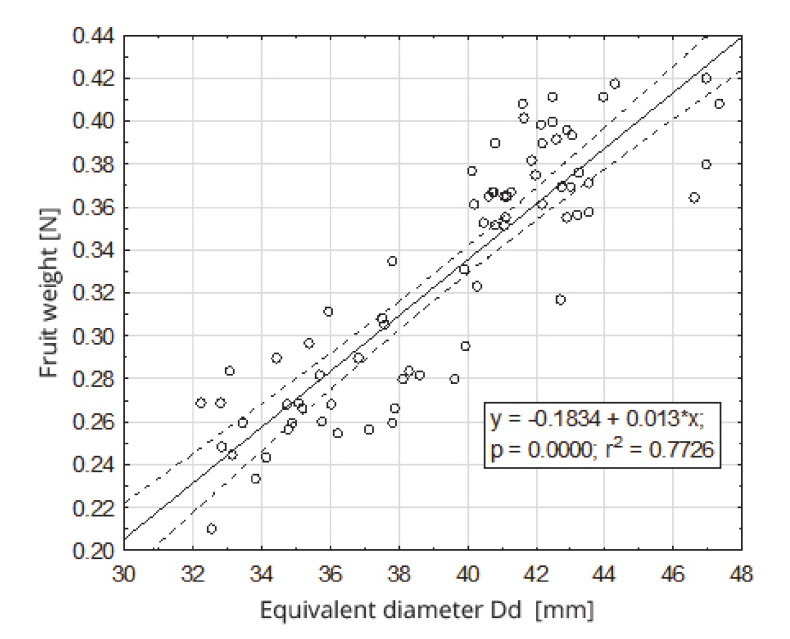
Dependence of the fruit weight on the equivalent diameter.

**Figure 7 sensors-20-04389-f007:**
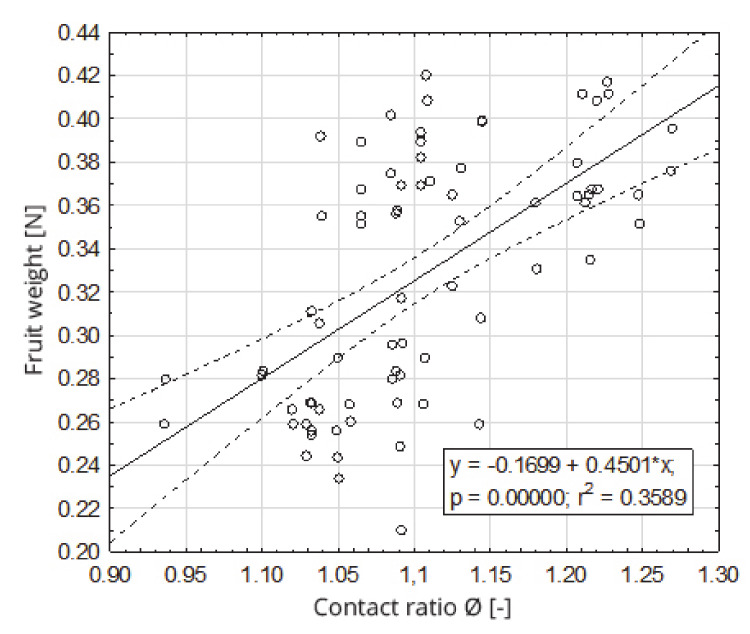
Dependence of the fruit weight on the sphericity coefficient.

**Figure 8 sensors-20-04389-f008:**
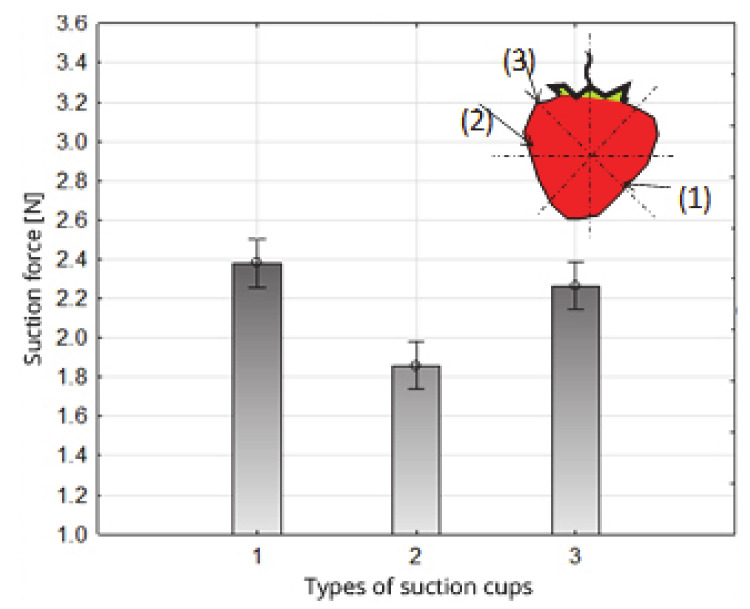
Impact of the type of suction cups on the value of the suction force of strawberry fruit.

**Figure 9 sensors-20-04389-f009:**
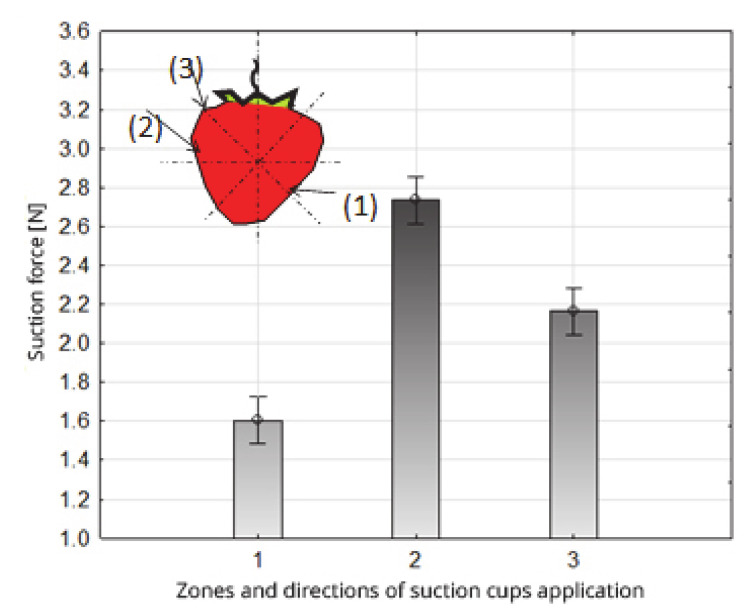
Impact of the zones and directions of suction cup application on the value of the suction force of the strawberry fruit.

**Figure 10 sensors-20-04389-f010:**
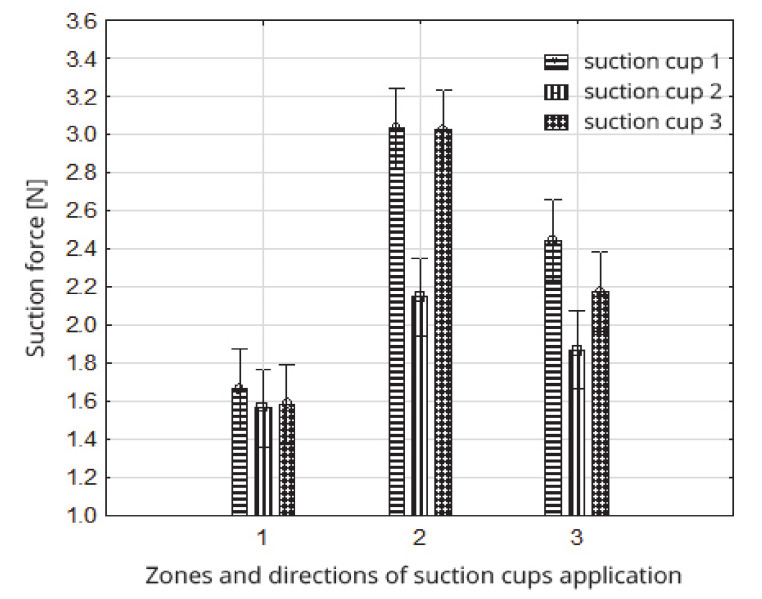
Impact of the zones and directions of suction cup application on the value of the strawberry fruit suction force within the type of suction cups.

**Figure 11 sensors-20-04389-f011:**
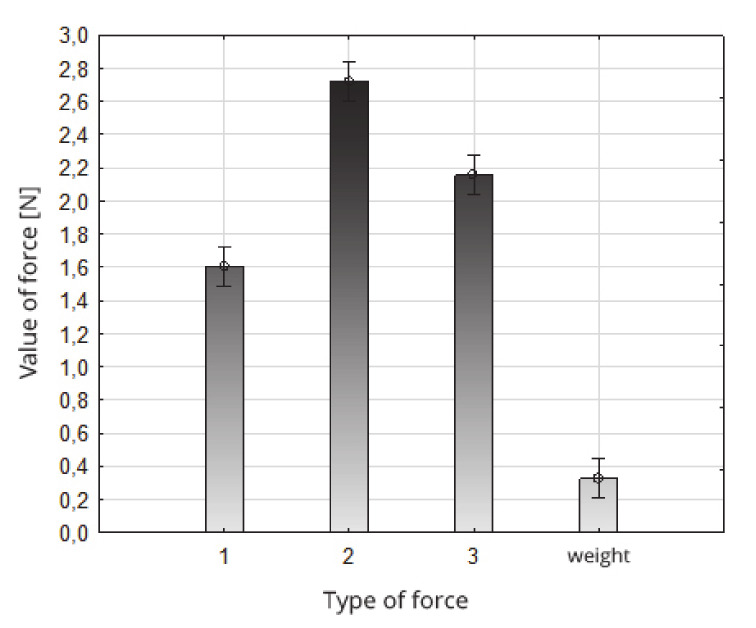
Comparison of strawberry fruit suction forces in zones and suction directions 1–3 with their weight.

**Figure 12 sensors-20-04389-f012:**
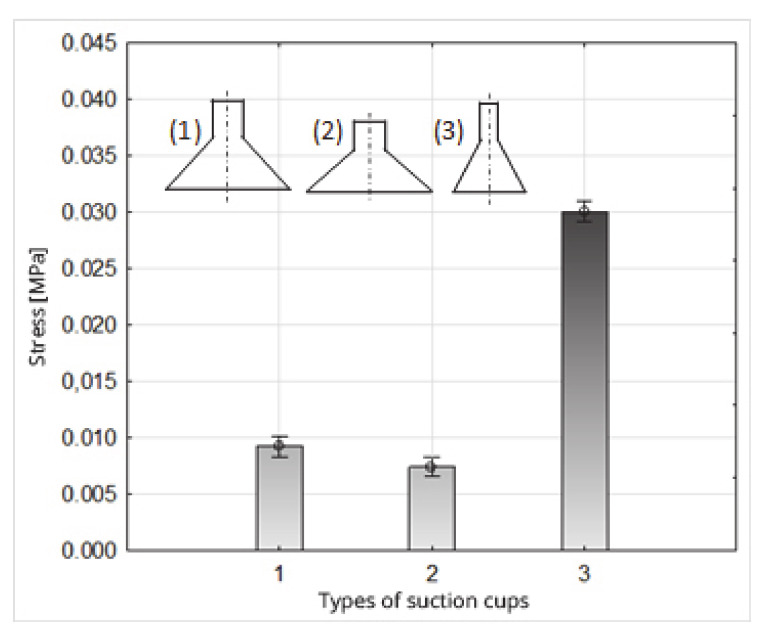
Influence of the type of suction cup on the stress value.

**Figure 13 sensors-20-04389-f013:**
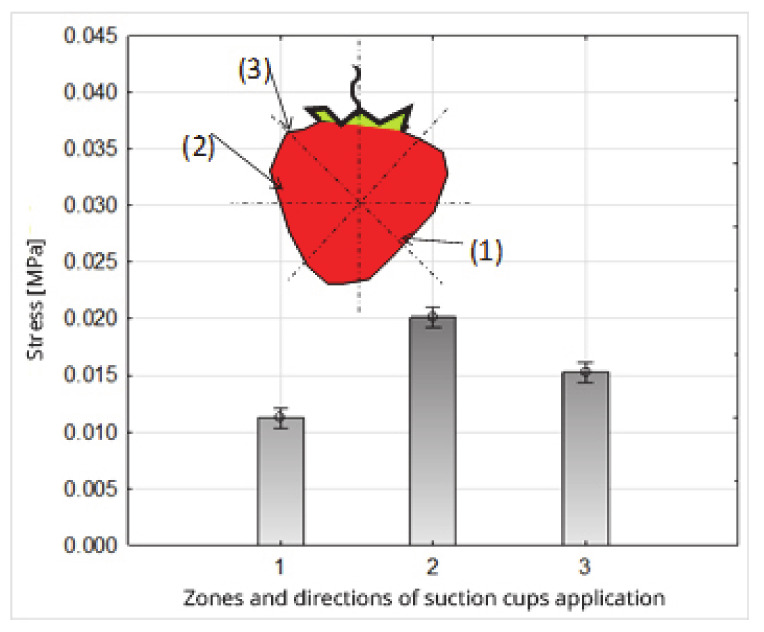
Influence of the zones and directions of the suction cup application on the stress value.

**Figure 14 sensors-20-04389-f014:**
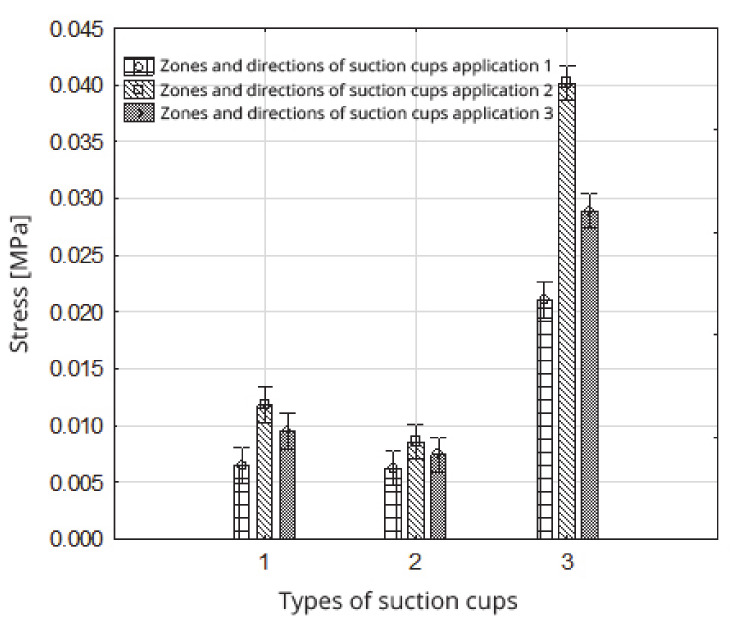
Impact of the type of suction cups within the zones and directions of suction cup application on the stress value.

**Figure 15 sensors-20-04389-f015:**
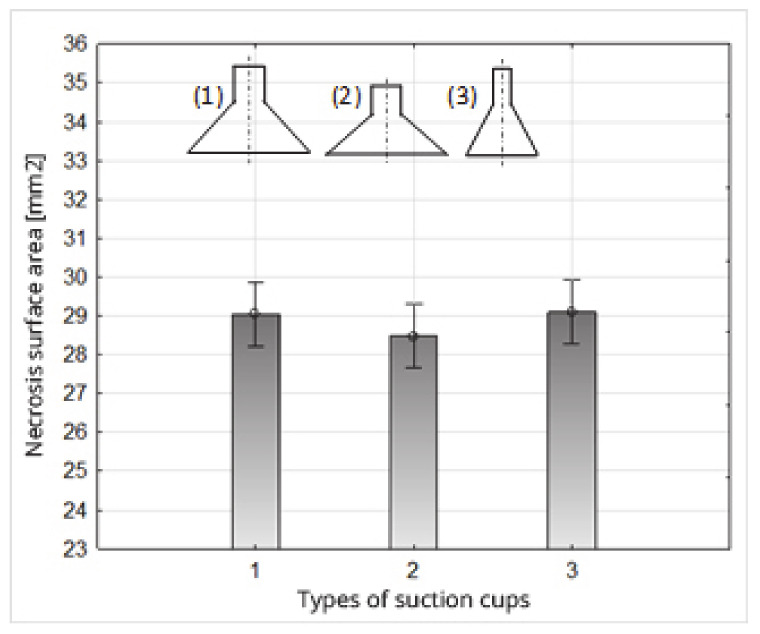
Impact of the type of suction cups on the necrosis surface area.

**Figure 16 sensors-20-04389-f016:**
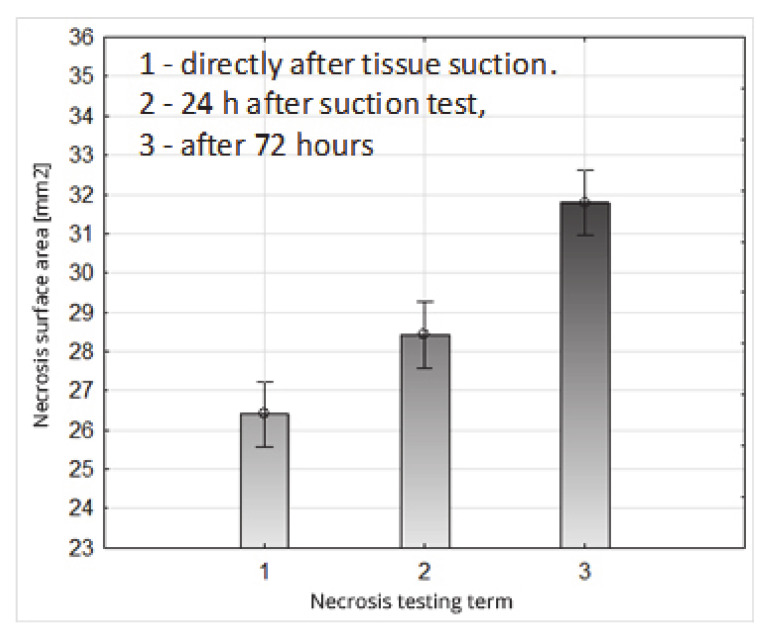
Impact of the necrosis testing term on their surface area.

**Figure 17 sensors-20-04389-f017:**
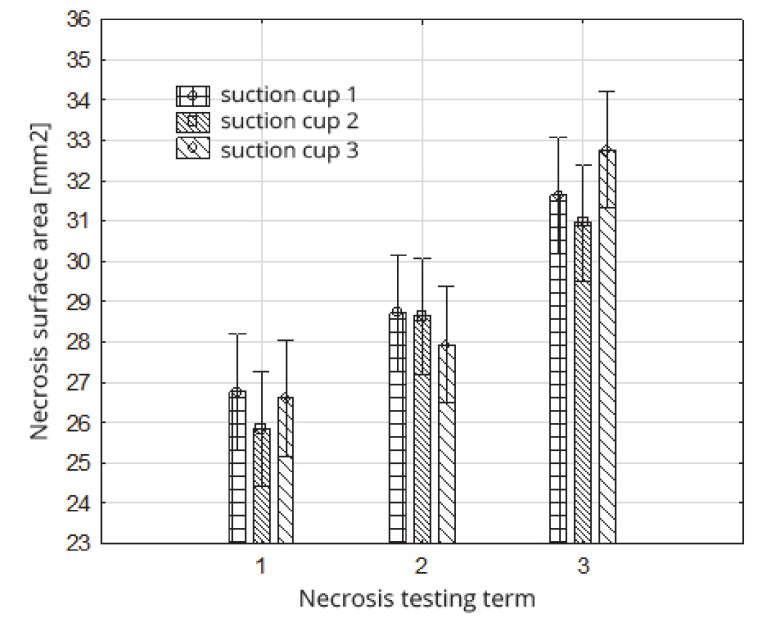
Impact of the necrosis testing term on the necrosis surface within the type of suction cups.

**Table 1 sensors-20-04389-t001:** Characteristics of the suction cups used.

Suction Cup Number	Geometric Dimensions of Suction Cup			Face Surfaces (A), mm^2^	Type of Suction Cup	Suction Cup Sketch
	d_in_, mm	d_out_, mm	h_s_, mm			
No 1	43	39	15	257.5	medium deep	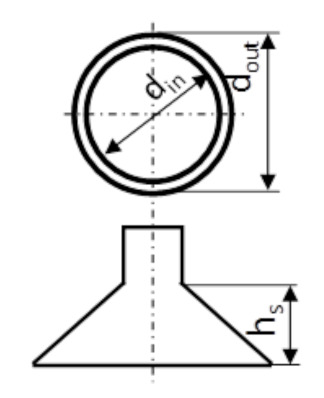
No 2	42	38	10	251.2	plate	
No 3	25	23	15	75.4	medium deep	

**Table 2 sensors-20-04389-t002:** Result of the analysis of variance with the effects of the interaction of the main factors—the impact of the type of suction cups, and the application zones and directions of the suction cups to the fruit—on the value of the suction force.

Qualitiv Predictor and Ineraction	Value of F-Snedecor Test	Probability Test
Intercept coefficient	3739.75	0.00
Types of suction cups {1}	20.08	0.00
Zones and directions of suction cups {2}	84.72	0.00
{1} x {2}	5.50	0.00

**Table 3 sensors-20-04389-t003:** The system of homogeneous groups (Duncan test); the effect of the type of suction cups on the value of the suction force.

Suction Force (N)		
Cups No 1	Cups No 2	Cups No 3
2.38 ^(1)^	1.86 ^(2)^	2.26 ^(1)^

^(.)^ Homogenous groups.

**Table 4 sensors-20-04389-t004:** System of homogeneous groups (Duncan test); the impact of zones and directions of suction cup application on the value of suction force.

Suction Force (N)		
Zones and Directions 1 for Applying Suction Cups	Zones and Directions 2 for Applying Suction Cups	Zones and Directions 3 for Applying Suction Cups
1.60 ^(1)^	2.72 ^(3)^	2.16 ^(2)^

^(..)^ Homogenous groups.

**Table 5 sensors-20-04389-t005:** System of homogeneous groups (Duncan test); influence of suction cup types, as well as zones and directions of suction cup application on the value of suction force.

Suction Force (N)								
Zones and Directions 1 for Applying Suction Cups			Zones and Directions 2 for Applying Suction Cups			Zones and Directions 3 for Applying Suction Cups		
Types of Suction Cups								
1	2	3	1	2	3	1	2	3
1.67 ^(1)^	1.56 ^(1)^	1.59 ^(1)^	3.03 ^(4)^	2.15 ^(2) (3)^	3.03 ^(4)^	2.44 ^(3)^	1.87 ^(1) (2)^	2.18 ^(2) (3)^

^(..)^ Homogenous groups.

**Table 6 sensors-20-04389-t006:** Result of the analysis of variance with the effects of the interaction of the major factors—the impact of the suction cup types, and the zones and directions of suction cup application on the value of stress.

Quality Predictor and Interaction	Value of F-Snedecor Test	Probability Test
Intercept Coefficient	3518.88	0.00
Types of Suction Cups {1}	768.97	0.00
Zones and Directions of Suction Cup Application {2}	96.62	0.00
{1}x{2}	33.43	0.00

**Table 7 sensors-20-04389-t007:** System of homogeneous groups (Duncan’s test); the effect of the type of suction cup on the stress value.

Stress (MPa)		
Cups No 1	Cups No 2	Cups No 3
0.0092 ^(2)^	0.0074 ^(1)^	0.0300 ^(3)^

^(..)^ Homogenous groups.

**Table 8 sensors-20-04389-t008:** System of homogeneous groups (Duncan test); the impact of the zones and directions of suction cup application on the stress value.

Stress (MPa)		
Zones and Directions 1 for Applying Suction Cups	Zones and Directions 2 for Applying Suction Cups	Zones and Directions 3 for Applying Suction Cups
0.0112 ^(1)^	0.0201 ^(3)^	0.0152 ^(2)^

^(..)^ Homogenous groups.

**Table 9 sensors-20-04389-t009:** System of homogeneous groups (Duncan test); the impact of the type of suction cups, as well as the zones and directions of the suction cup application on the value of stress.

Stress (MPa)								
Zones and Directions 1 for Applying Suction Cups			Zones and Directions 2 for Applying Suction Cups			Zones and Directions 3 for Applying Suction Cups		
Types of Suction Cups								
No 1	No 2	No 3	No 1	No 2	No 3	No 1	No 2	No 3
0.0065 ^(1)^	0.0062 ^(1)^	0.0210 ^(4)^	0.0118 ^(3)^	0.0085 ^(1) (2)^	0.0402 ^(6)^	0.0095 ^(2)^	0.0074 ^(1) (2)^	0.0289 ^(5)^

^(..)^ Homogenous groups.

**Table 10 sensors-20-04389-t010:** Result of the analysis of variance with the effects of the interaction of the main factors—the impact of the type of suction cups and the time of necrosis examination on the necrosis surface.

Qualitiy Predictor and Ineraction	Value of F-Snedecor Test	Probability Test
Intercept Coefficient	14037.54	0.00
Time of Necrosis Examination {1}	41.55	0.00
Types of Suction Cups	0.65	0.52
{1}x{2}	0.86	0.49

**Table 11 sensors-20-04389-t011:** System of homogeneous groups (Duncan’s test); the impact of the necrosis test date on the necrosis surface.

Necrosis Surface Area (mm^2^)		
Necrosis Testing Term 1	Necrosis Testing Term 2	Necrosis Testing Term 3
26.40 ^(1)^	28.43 ^(2)^	31.79 ^(3)^

^(..)^ Homogenous groups.

**Table 12 sensors-20-04389-t012:** Examples of necrosis and the average surface area values of their occurrence.

Type of Sunction Cups	Average Value Of The Area Formed Necrosis (mm)			Examples of Necrosis		
	0 (h)	24 (h)	72 (h)	0 (h)	24 (h)	72 (h)
1	26.76	28.72	31.64	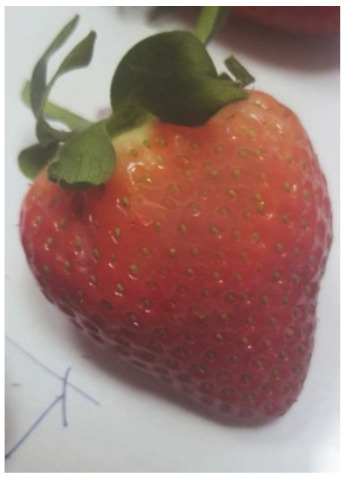	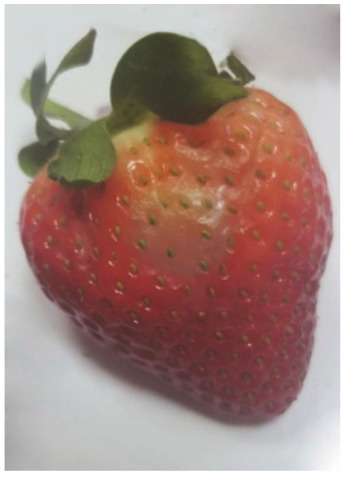	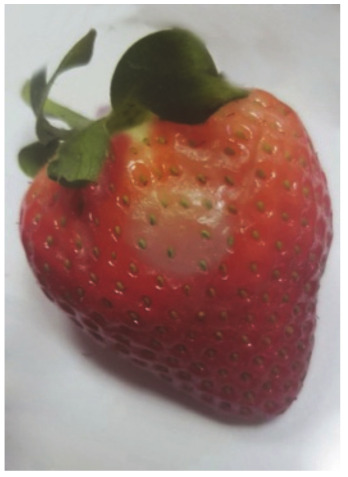
2	25.84	28.64	30.96	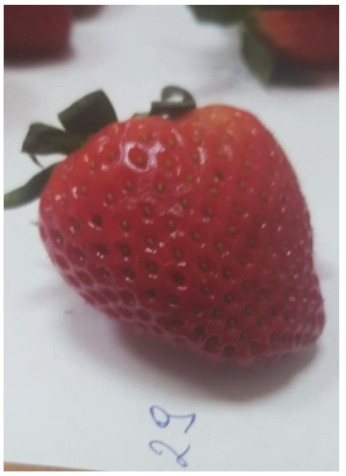	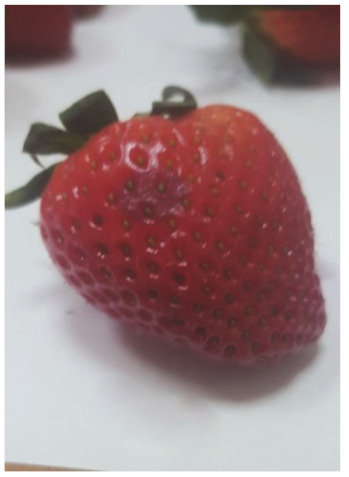	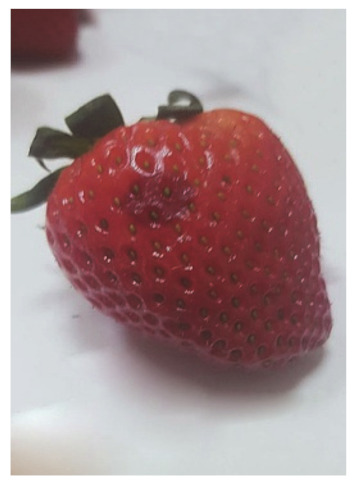
3	26.60	27.92	32.76	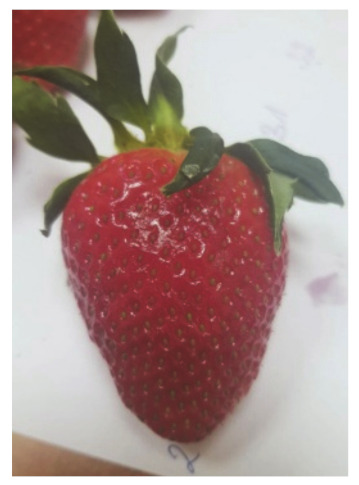	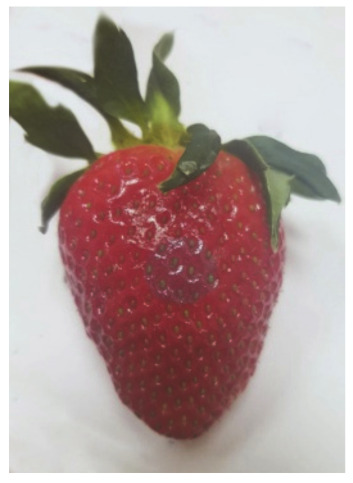	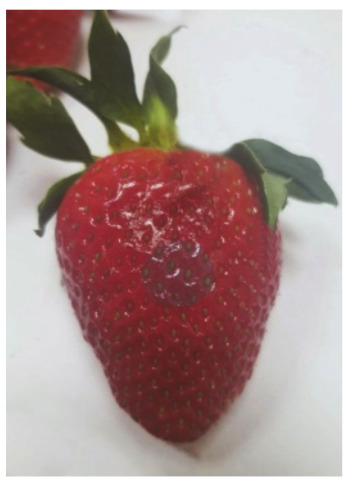

**Table 13 sensors-20-04389-t013:** System of homogeneous groups (Duncan’s test); impact of the necrosis testing term and the suction cup type on the necrosis surface.

Necrosis Surface Area (mm^2^)								
Necrosis Testing Term 1			Necrosis Testing Term 2			Necrosis Testing Term 3		
Type of Sunction Cups								
No 1	No 2	No 3	No 1	No 2	No 3	No 1	No 2	No 3
26.76 ^(1) (2)^	25.84 ^(2)^	26.60 ^(1) (2)^	28.72 ^(1)^	28.64 ^(1)^	27.92 ^(1) (2)^	31.64 ^(3)^	30.96 ^(3)^	32.76 ^(3)^

^(..)^ Homogenous groups.

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
