# Peer review of "Analysis of the Pneumatic System Parameters of the Suction Cup Integrated with the Head for Harvesting Strawberry Fruit"

_sensors, 2020, doi:10.3390/s20164389_

Round 1

Reviewer 1 Report

This is a study to determine the influence of sucking cups for strawberries, which will be quite beneficial for robotic harvesting, which is a subject of increasing demand.

There are a number of comments to be addressed, which are as follow:

Line 91: In the strawberry harvesting literature, please also review the recent paper on Sensors, as it has reviewed the processing time for strawbeery harvesting in comparison to other fruits: “Agricultural Robotics for Field Operations, Sensors 20(9):2672, 2020.

Lines 92-180: The state of the art on strawberry robotic systems is quite extensive. You should try to make it shorten trying to synthesize the results of the papers and not explain them into high detail.

Line 255: Figure 1 is important, but it is not very attractive. Please try to put some colours and images of the components, i.e. computer.

Line 264: Please explain how you decided to use the three suction cups for the tests.

Line 283: Please explain further why you decided to use these morphological features for the strawberry fruit.

Line 302: You have to use past tense when you describe experiments that have taken place. Please follow this throughout the paper.

Line 301-302: Please explain how you calculated the necrosis using LabView.

Line 313: Change STSTISTICA to STATISTICA

Line 392: Table 3 on the system of homogeneous groups is not clear and has to be redone to be more informative, as well as Table 4, 5, 9 and 11. Please try to find another way to present these Tables.

Line 553 on Summary, change it to Discussion.

Author Response

I hope the version you submitted will be acceptable.

All changes were made in the review mode.

Kindly regards.

Reviewer 2 Report

  1. This work has a good contribution to the design of end effectors using suction cup for strawberry harvest robot.
  1. There is a typo in the paper title, “sunction cup” should be“suction cup”.
  1. The content of “Introduction” is too lengthy. It counts for 209 lines (from Line 33 to Line 241) in total, and it takes a proportion of 34.8% among the whole paper (598 lines). In fact, much of the contents in “Introduction” is NOT closely related to the main objective of this paper, a big reduction of contents in “Introduction” is strongly suggested.
  1. “Test Results” (Line 318 to 552) is weak in discussion, more elaboration in comprehensive analyses in depth is expected for revision.
  1. The way of presentation of this manuscript seems like a report instead of a journal paper. Please revise.

Author Response

(The authors gave the same response as above.)

Round 2

Reviewer 1 Report

The authors have successfuly addressed the reviewer's comments and the paper can be accepted.

Reviewer 2 Report

The manuscript has been well revised. However, there is a typo in the paper title, “sunction cup” should be “suction cup”. After correct the this typo, the paper can be accepted.